# Condensation of 4-Tert-butyl-2,6-dimethylbenzenesulfonamide with Glyoxal and Reaction Features: A New Process for Symmetric and Asymmetric Aromatic Sulfones

**DOI:** 10.3390/molecules27227793

**Published:** 2022-11-12

**Authors:** Artyom E. Paromov, Sergey V. Sysolyatin, Irina A. Shchurova

**Affiliations:** Laboratory for Chemistry of Nitrogen Compounds, Institute for Problems of Chemical and Energetic Technologies, Siberian Branch of the Russian Academy of Sciences (IPCET SB RAS), Biysk 659322, Russia

**Keywords:** condensation, domino reactions, aromatic sulfonamides, aromatic sulfones, aromatic sulfanes, aromatic disulfanes, 1,2-hydride shift, 1,3,5-triazinane, 2,4,6,8,10,12-hexaazatetracyclo[5.5.0.0^3,11^.0^5,9^]dodecane

## Abstract

The synthesis of substituted aza- and oxaazaisowurtzitanes via direct condensation is challenging. The selection of starting ammonia derivatives is very limited. The important step in developing alternative synthetic routes to these compounds is a detailed study on their formation process. Here, we explored an acid-catalyzed condensation between 4-tert-butyl-2,6-dimethylbenzenesulfonamide and glyoxal in aqueous H_2_SO_4_, aqueous acetonitrile and acetone, and established some new processes hindering the condensation. In particular, an irreversible rearrangement of the condensation intermediate was found to proceed and be accompanied by the 1,2-hydride shift and by the formation of symmetric disulfanes and sulfanes. It has been shown for the first time that aldehydes may act as a reducing agent when disulfanes are generated from aromatic sulfonamides, as is experimentally proved. The condensation between 4-tert-butyl-2,6-dimethylbenzenesulfonamide and formaldehyde resulted in 1,3,5-tris((4-(tert-butyl)-2,6-dimethylphenyl)sulfonyl)-1,3,5-triazinane. It was examined if diimine could be synthesized from 4-tert-butyl-2,6-dimethylbenzenesulfonamide and glyoxal by the most common synthetic procedures for structurally similar imines. It has been discovered for the first time that the Friedel–Crafts reaction takes place between sulfonamide and the aromatic compound. A new synthetic strategy has been suggested herein that can reduce the stages in the synthesis of in-demand organic compounds of symmetric and asymmetric aromatic sulfones via the Brønsted acid-catalyzed Friedel–Crafts reaction, starting from aromatic sulfonamides and arenes activated towards an electrophilic attack.

## 1. Introduction

The development of defense technology is inseparably linked to the design of new high-energy compounds that are superior in energy-mass and performance characteristics to the existing ones. Typically, cyclic nitramines such as hexahydro-1,3,5-trinitro-1,3,5-triazine (RDX) and 1,3,5,7-tetranitro-1,3,5,7-tetrazocine (HMX) serve as the reference explosives (Figure 1). These explosives are well-studied and widely used in the civil and defense industries [1,2].

The important milestone in advancing the chemistry of high-energy materials is the discovery of caged nitramines such as *N*-polynitro-substituted aza- and oxaazaisowurtzitanes. Transiting from the cyclic to the strained cage structure allows the energy-mass characteristics of these compounds to be enhanced considerably.

2,4,6,8,10,12-Hexanitro-2,4,6,8,10,12-hexaazatetracyclo[5.5.0.0^3,11^.0^5,9^]dodecane (hexanitrohexaazaisowurtzitane, HNIW, CL-20) (Figure 1) exhibits one of the highest detonation rates among all of the explosives (V_0_*D* = 9.36 (ε) km/s), the highest density among the known nitramines (ρ = 2.044 g/cm^3^) [3,4,5,6], a positive heat of formation (ΔH*f* = 454 kJ/mol), and optimum oxygen balance (−11.0) and detonation pressure (420 kbar) [7,8,9,10,11,12]. HNIW excels the other high-energy explosives such as HMX, RDX, PETN, etc. CL-20 is touted as a component of advanced solid propellants [13,14,15,16,17,18] and composite explosives [19,20,21,22,23,24,25,26,27,28,29,30,31,32,33,34].

Replacing some nitramine groups in the CL-20 cage by ester units has been found to reduce the sensitivity of the resulting explosives, without a substantial density loss. 4,10-Dinitro-2,6,8,12-tetraoxa-4,10-diazatetracyclo[5.5.0.0^3,11^.0^5,9^]dodecane (TEX) is an insensitive explosive comprising two nitramine moieties and four ester bridges, and has a high density (ρ = 2.014 g/cm^3^) and a high detonation velocity (V_0_*D* = 8.66 km/s, calcd.) (Figure 1) [35,36]. In most cases, the TEX-based explosive formulations are more powerful than composite explosives based on 3-nitro-1,2,4-triazol-5-one (NTO) [37].

Since the discovery of *N*-polynitro-substituted aza- and oxaazaisowurtzitanes in the mid-80s and to present, the synthetic technologies for these compounds have been developed and upgraded worldwide; however, their production cost remains high so far, limiting their use for large-scale applications.

The most promising and cost-effective synthetic method towards *N*-polynitro-substituted aza- and oxaazaisowurtzitanes is the direct condensation between ammonia derivatives and glyoxal, followed by the exhaustive nitration of the resulting aza- and oxaazaisowurtzitanes.

At this point, the formation mechanism of the hexaazaisowurtzitane cage is underexplored. In the overwhelming majority of cases, the hexaazaisowurtzitane cage originates from the condensation of ammonia derivatives containing an aromatic moiety or a multiple bond linked to the amino group via the methylene bridge [7,38,39]. The reasons behind it having such a high selectivity towards the structure of ammonia derivatives are unknown. There is, therefore, no doubt that the most important step in designing alternative synthetic routes to *N*-polynitro-substituted aza- and oxaazaisowurtzitanes is an in-depth study into that process in order to identify new formation mechanisms towards aza- and oxaazaisowurtzitane cages, in particular, the structure–property relationship for the starting ammonia derivatives and their capability of being condensed with glyoxal to furnish hexaazaisowurtzitane derivatives.

Here, we reported the findings of the condensation process between the high-basicity substituted sulfonamide, 4-tert-butyl-2,6-dimethylbenzenesulfonamide (**1**), and glyoxal in a ratio of 2:1 to synthesize aza- and oxaazaisowurtzitane derivatives.

## 2. Results and Discussion

We have previously investigated the condensation of substituted sulfonamides (methanesulfonamide, benzenesulfonamide, 4-dimethylaminobenzenesulfonamide or isopropylsulfonamide) with aromatic carboxamides (benzamide) and glyoxal [40,41,42,43].

The detailed study on the condensation between benzamide and glyoxal in a ratio of 2:1 in different polar protic and aprotic solvents revealed that the carbamide group in the benzamide molecule undergoes hydrolysis under the glyoxal condensation conditions, and the condensation products undergo intramolecular transpositions (1,2-hydride shift; cyclization). No formation of caged compounds was documented. In view of this, we concluded that aromatic carboxamides (probably all carboxamides) are of little utility as the scaffold for the synthesis of aza- and oxaazaisowurtzitanes via direct condensation [43].

We earlier synthesized a range of new oxaazaisowurtzitane derivatives by the condensation between substituted sulfonamides and glyoxal, discovered some polyheterocyclic caged systems, and established new condensation patterns [40,41,42,43]. In particular, the increased basicity of the amido group in the sulfonamide molecule was found to facilitate the incorporation of aza groups into the oxaazaisowurtzitane cage. Unlike benzenesulfonamide, 4-dimethylaminobenzenesulfonamide was condensable with glyoxal to yield an oxaazaisowurtzitane derivative bearing three aza groups. In addition, the condensation of 4-dimethylaminobenzenesulfonamide furnished a byproduct that may be suggestive of the formation of a dioxatetraazaisowurtzitane derivative [42].

In the present study, we chose the high-basicity substituted sulfonamide **1** as a substrate for the study. In contrast to 4-dimethylaminobenzenesulfonamide, its donor substituents are not liable to protonation and cannot lead to a decreased basicity of sulfonamide in highly acidic media. The chemistry of sulfonamide **1** is almost understudied. We have not managed to find data on its condensation with aldehydes.

The condensation between sulfonamide **1** was examined in aqueous H_2_SO_4_, aqueous acetonitrile, and acetone over an H_2_SO_4_ catalyst. The reaction mixture was diluted with water and extracted with ethyl acetate. The extract was analyzed by HPLC. 

We also examined if diimine could be formed from sulfonamide **1** and glyoxal under the conditions mostly used for the synthesis of structurally similar imines. The compound is of interest as the scaffold for the synthesis of the respective hexaazaisowurtzitane derivative via trimerization. The presumed trimerization mechanism is described in [38].

First, we looked into the condensation between sulfonamide **1** and glyoxal in aqueous H_2_SO_4_. The reactions were carried out at room temperature for 2 h. Because sulfonamide **1** is poorly soluble in water, it was dissolved in aqueous H_2_SO_4_. The minimum content of H_2_SO_4_ required for the dissolution was 73% in the mixture. The enhanced solubility of the amide appears to be due to protonation. The sulfonamide was dissolved in aqueous H_2_SO_4_ first, and glyoxal was then added portion-wise to the mixture with vigorous stirring at room temperature for 1–2 min. All of the experiments used 0.3 g of sulfonamide and 0.5 mL of water. The condensation was studied at an H_2_SO_4_ content of 73% to 83% in the mixture.

The complex mixed compounds resulting from the condensation were separated by preparative chromatography. The major reaction products isolated were 2-((4-(tert-butyl)-2,6-dimethylphenyl)sulfonamido)-N-((4-(tert-butyl)-2,6-dimethylphenyl)sulfonyl)acetamide (**2**), 1,2-bis(4-(tert-butyl)-2,6-dimethylphenyl)disulfane (**3**) and bis(4-(tert-butyl)-2,6-dimethylphenyl)sulfane (**4**).

Compound **2** was likely formed by the 1,2-hydride shift in the molecule of 1,2-bis((4-(tert-butyl)-2,6-dimethylphenyl)sulfonamido)-1,2-ethanediol (**5**) (Figure 1). This possible transposition was first established in our previous study [43]. Thus, this process proceeds with both carboxamides and sulfonamides. The formation of diol **5** was explicitly corroborated by the formation of compound **2**. The diol appears to have a low stability under the reaction conditions and undergoes transposition immediately.

In the experiments with aqueous H_2_SO_4_, sulfonamide **1** was hydrolyzed to 4-(tert-butyl)-2,6-dimethylbenzenesulfonic acid (**6**), which was desulfated to 1-(tert-butyl)-3,5-dimethylbenzene (**7**) (Figure 2). Acid **6** is highly soluble in water; therefore, its content in the extract assayed by HPLC does not reflect the accurate quantity. Once extracted, some of the acid remained in the aqueous phase.

To identify acid **6**, the counter synthesis thereof was effected by the procedure reported in [44]. 

The attempt to isolate acid **6** by extraction with ethyl ether acidified with HCl appeared to furnish protonated acid **6(H^+^)** monohydrate as crystals (Figure 3). The stability of this compound comes from the charge delocalization in the aromatic system. The broad singlet of two acid protons in the ^1^H NMR was at 9.58. The elemental analysis data suggest that the compound was isolated in the hydrated form. Good water solubility can be distinguished among the properties of this compound.

Since the thiols are air-oxidizable to disulfane with no catalysts [45,46], we hypothesized that disulfane **3** is formed by the interaction of two molecules of 4-tert-butyl-2,6-dimethylbenzenethiol (**8**) in the presence of atmospheric oxygen (Figure 4).

In that case, thiol **8** was generated by the reduction of acid **6** with glyoxal, as was experimentally confirmed. Holding amide **1** in 80% H_2_SO_4_ in the absence of glyoxal did not furnish disulfane **3** and sulfane **4**. Replacing glyoxal by formaldehyde (a sulfonamide-to-formaldehyde molar ratio of 1:1) resulted in a small amount of sulfane **4** (0.4% in the extract (HPLC)); reaction time was 2 h; the H_2_SO_4_ content in the mixture was 80% with trace amounts of compound **3**. When formaldehyde was added portion-wise, 1,3,5-tris((4-(tert-butyl)-2,6-dimethylphenyl)sulfonyl)-1,3,5-triazinane (**9**) precipitated abundantly (Figure 5). Most of the formaldehyde left the reaction mixture at the outset of the synthesis. 1,3,5-Triazinane **9** was obtained in a 50.5% yield (no optimization of the synthetic process was performed). We have not managed to find information on the use of aldehydes as reducing agents of substituted sulfoacids.

Sulfane **4** was likely generated by the Friedel–Crafts reaction through the S_E_Ar mechanism to give an intermediate arene σ-complex **10** (Figure 6). The (4-tert-butyl-2,6-dimethylphenyl)sulfonium (**11^+^**) cation attacked compound **7** activated at the 4th position. The stability of cation **11^+^** is explained by the charge delocalization in the aromatic system activated by donor substituents. H^+^ acted as the Lewis acid in that process.

Table 1 lists major reaction products in the extract after sulfonamide **1** reacted with glyoxal in a ratio of 2:1 in H_2_SO_4_ of varied concentrations.

Several conclusions can be made from the data outlined in Table 1. For instance, the starting sulfonamide **1** was engaged most actively in the condensation reaction when the H_2_SO_4_ content in the mixture was 76% (Table 1, Entry 2), the major reaction product being compound **2**.

As the H_2_SO_4_ content in the mixture was raised from 76% (Table 1, Entry 2) to 83% (Table 1, Entry 8), the content of sulfonamide **1** increased, and the content of compound **2** sharply decreased (in the extract) through to its complete absence when the H_2_SO_4_ content in the mixture was 81% (Table 1, Entry 6), which was likely due to the activated hydrolysis of the condensation products.

Over the entire acidity range in question, amide **1** was observed to be hydrolyzed to acid **6,** and the latter was desulfated to compound **7**, in which case the hydrolysis rate was increasing up to the H_2_SO_4_ content of 79% in the mixture for acid **6** (Table 1, Entries 1–4) and up to the H_2_SO_4_ content of 81% for compound **7** (Table 1, Entries 1–6) as the reaction mixture acidity was raised. The H_2_SO_4_ content above 77% in the mixture resulted in sulfane **4** (Table 1, Entries 3–8). The content of **4** in the reaction products was rising up to the H_2_SO_4_ content of 82% in the mixture. The increase in sulfane **4** in the reaction mixture concurrently with a decline in compounds **3** and **7** is on a par with the suggested formation mechanism of sulfane **4** (Figure 6; Table 1, Entries 3–8).

The reactions performed in pure H_2_SO_4_ with no extra water afforded the highest yields of compounds **2** and **4**. The highest amount of compound **2** (33% in the mixture (HPLC)) was achieved when the condensation was conducted in a small amount of H_2_SO_4_ (31.5% in the mixture) at room temperature for 11 h. Compound **4** was best formed (32% in the mixture (HPLC)) when the condensation was carried out in H_2_SO_4_ (64% in the mixture) for 4 h at 60 °C. In both cases, glyoxal was added portion-wise to the mixture for 1–2 min.

Further, we examined the condensation between sulfonamide **1** and glyoxal in aqueous acetonitrile. Water addition to the mixture was required to enhance the solubility of glyoxal (40%) and prevent it from precipitation as a tarry low-solubility sediment. The reactions were carried out at 30 °C for 5 h. The condensation was investigated by varying the H_2_SO_4_ content from 1% to 63% in the mixture.

The condensation of sulfonamide **1** with glyoxal took place actively when the H_2_SO_4_ content was 30–63%, resulting in a large number of products, among which the major one was *N*,*N*’-(1,2-bis((4-(tert-butyl)-2,6-dimethylphenyl)sulfonamido)ethane-1,2-diyl)diacetamide (**12**) (Figure 7). The highest content of 18% of this compound in the extract was obtained when the H_2_SO_4_ content was 63%. Compound **12** resulted most probably from the condensation between acetamide and diol **5**. In this case, acetamide originated from the acid-catalyzed hydrolysis of acetonitrile. The generation of compound **12** corroborates that diol **5** originates also from the condensation between sulfonamide **1** and glyoxal.

The condensation between sulfonamide **1** and glyoxal in acetone (no extra water added) led to a large number of products, among which the major one was a tarry product from the reaction with acetone. The highest content of 16% of the compound in the extract was obtained when the H_2_SO_4_ content was 24%. The ^13^C NMR spectrum of the compound had signals at 208.4 (C=O), 54.9 (CH_2_), 53.6 (CH_2_) and 23.2 (CH_3_), as well as signals typical of the aromatic system at 154.6 (C), 138.1 (C), 136.6 (C) and 128.3 (CH). It was impossible to resolve the structure of the compound.

We further examined the condensation between sulfonamide **1** and glyoxal in order to obtain the respective diimine. The reaction was conducted under conditions used for the synthesis of structurally similar imines [47,48,49,50,51,52,53]. Glyoxal was dewatered by distillation of the water–toluene azeotrope. The content of the reaction products was quantified by HPLC.

The reaction almost did not proceed in methylene chloride or chloroform at reflux in the presence of Et_3_N and excess TiCl_4_, in aqueous formic acid in the presence of sodium benzenesulfinate at room temperature, and in methylene chloride or dichloroethane in the presence of pyrrolidine (10% in the mixture) and 4 Å molecular sieves (1 g/mmol).

Refluxing sulfonamide **1** with dewatered glyoxal in a 16% toluene solution of titanium (IV) isopropoxide for 8 h furnished disulfane **3** (6% in the mixture (HPLC)). The reaction performed in excess pure titanium (IV) isopropoxide at 160 °C for 12 h increased the content of disulfane **3** from 6% to 24% in the mixture. Glyoxal in this reaction acted as a reducing agent. Compound **3** was not formed in the absence of glyoxal. 

Refluxing sulfonamide **1** with dewatered glyoxal in boiling toluene over a boron trifluoride etherate (2.3% in the mixture) for 11 h gave compound **2** (16% in the mixture (HPLC)), disulfane **3** (1.9% in the mixture (HPLC)), and **7** (13.3% in the mixture (HPLC)). The reaction over a PTSA catalyst (1.3% in the mixture) for 11 h produced compound **2** (7% in the mixture (HPLC)). The reaction over a TfOH catalyst (2.2% in the mixture) for 2 h produced compound **7** (7% in the mixture (HPLC)) and 5-(tert-butyl)-1,3-dimethyl-2-tosylbenzene (**13**) (70% in the mixture (HPLC)) (Figure 8). The reaction performed in 1,2-DCE at reflux over a TfOH catalyst (3% in the mixture) for 2 h delivered 2,2′-sulfonylbis(5-(tert-butyl)-1,3-dimethylbenzene) (**14**) (50% in the mixture (HPLC)) and sulfane **4** (11% in the mixture (HPLC)) (Figure 8).

We believe that compounds **13** and **14**, as well as sulfane **4**, originated from the Friedel–Crafts reaction via the S_E_Ar mechanism illustrated in Figure 6. In the case of compound **13**, cation **15^+^** electrophilically attacked toluene, while in the case of compound **14**, cation **15^+^** electrophilically attacked compound **7**. The existence of cation **15^+^** is corroborated by numerous studies on the synthesis of analogous compounds from sulfoacid chloroanhydrides and aromatics in the presence of Lewis acids [54,55,56]. The stability of cation **15^+^**, as well as of **11^+^**, is explained by the charge delocalization in the aromatic system activated by donor substituents. H^+^ acted as the Lewis acid. The formation of cation **15^+^** in that case may be due to the detachment of the ammonia molecule from the protonated sulfonamide molecule **1(H^+^)** or due to the water detachment from the protonated sulfoacid **6(H^+^)** (Figure 8). Because the reaction proceeded in zeolite-predewatered solvents at reflux that give an azeotropic mixture with water, the first option taking place to cleave the C-N is the most probable.

Glyoxal was not involved in the formation of compounds **13** and **14**, as is experimentally proved. Refluxing sulfonamide **1** in toluene over the TfOH catalyst (3% in the mixture) for 5 h also furnished compound **13** (80% in the mixture (HPLC)). Refluxing sulfonamide **1** in 1,2-DCE over the TfOH catalyst (3% in the mixture) for 5 h resulted also in compound **14** (68% in the mixture (HPLC)). The reaction in toluene was slower than in 1,2-DCE and was incomplete, which is likely due to TfOH reacting with toluene and escaping the reaction mixture. Since the synthesis of the sulfones was out of the scope of the present study, no optimization of the process for compounds **13** and **14** was performed.

Sulfones have been known for long and are widely applied in the synthesis of polymers [57,58,59,60], bioactive compounds [61,62], agrochemicals [63,64], fluorescent compounds [65] and other organics. Despite this, we have not managed to find reports on the synthesis of aromatic sulfones by the Brønsted acid-catalyzed Friedel–Crafts reaction in which aromatic sulfonamides and aromatics are utilized as the starting reactants. The synthetic process discovered herein for symmetric and asymmetric sulfones can be useful as an alternative to the common synthetic methods. This process can significantly shorten the stages in the synthesis of in-demand organic compounds [57,58,59,60,61,62,63,64,65], is easy to perform, and provides a good yield. The basic requirement for this process is probably the use of aromatic sulfonamides highly activated by donor substituents. Additionally, this process can probably be used for the synthesis of sulfones from aromatic sulfoacids highly activated by donor substituents.

## 3. Materials and Methods

The reagents were purchased from commercial sources and used as received, unless mentioned otherwise. Commercially available compounds were used without further purification, unless stated otherwise. Melting points were determined on a Stuart SMP30 melting point apparatus (Bibby Scientific Ltd., Staffordshire, UK). Infrared (IR) spectra were recorded on a Simex FT-801 Fourier transform infrared spectrometer (Simex, Novosibirsk, Russia) in KBr pellets or in a liquid film. ^1^H and ^13^C NMR spectra were recorded on a Bruker AV-400 instrument (Bruker Corporation, Billerica, MA, USA) at 400 MHz and 100 MHz. Chemical shifts are expressed in ppm (δ). Elemental analysis was performed on a ThermoFisher FlashEA 1112 elemental analyzer (ThermoFisher, Waltham, MA, USA). For preparative chromatography, silica gel Kieselgel 60 (0.063–0.2 mm, Macherey-Nagel GmbH & Co. KG, Dueren, Germany) was used. HPLC analysis was performed on an Agilent 1200 chromatograph (Agilent Technologies, Waldbronn, Germany) with a diode array detector. The separation was carried out on a Hypersil ODS (100 × 2.1 mm, a 5 µm mesh) column. Mixed solvents A (0.2% phosphoric acid) and B (acetonitrile) were used as the mobile phase. The mobile phase composition was varied in the gradient mode: the concentration of solvent B was linearly raised from 45 to 100% for 25 min and maintained at this level for another 25 min. The flowrate of the eluent was 0.25 mL/min, the column temperature was 25 °C, detection was run at a 210 nm wavelength, and the sample volume was 3 µL. The column conditioning time between successive injections was 15 min.

## 4. Experimental Section

### 4.1. Synthesis of 2-((4-(Tert-butyl)-2,6-dimethylphenyl)sulfonamido)-N-((4-(tert-butyl)-2,6-dimethylphenyl)sulfonyl)acetamide (***2***)

Sulfonamide **1** (3 g, 0.012 mol) was dissolved in H_2_SO_4_ (2 mL, 94%) at room temperature. Glyoxal (0.902 g, 40%, 0.006 mol) was then added portion-wise to the mixture with stirring at 22–24 °C for 3–4 min. During the portion-wise addition, a great quantity of a tarry sediment precipitated. The whole was further periodically stirred manually once an hour for 11 h. Upon completion of the time, the reaction mixture was poured over with water (20 mL) and extracted with ethyl acetate. The extract was washed with water and brine, and dried over Na_2_SO_4_ and evaporated to dryness on a rotary evaporator in vacuo (50 °C bath temperature). The residue was subjected to preparative chromatography. Mixed chloroform and glacial acetic acid in a volume ratio of 10:0.5 were used as the eluent. Fractions with Rf = 0.48 were collected. The solvents were removed to dryness from the collected fractions in vacuo by a rotary evaporator (50 °C bath temperature). The residue was recrystallized from diethyl ether and dried at room temperature until constant weight to furnish compound **2** as a white crystalline powder.

Yield: 0.696 g (98% assay (HPLC)), 1.305 mmol (21% calculated as compound **1**). Mp = 203–205 °C (Et_2_O; dec.). IR (KBr): ν = 3397, 3271, 3061, 2964, 2905, 2869, 1728, 1697, 1594, 1557, 1443, 1407, 1331, 1227, 1193, 1174, 1145, 1112, 1049, 928, 869, 849, 700, 645 cm^−1^. ^1^H NMR (CDCl_3_): δ = 1.32 (s, 18H), 2.64 (s, 6H), 2.73 (s, 6H), 3.58 (d, J = 6.08 Hz, 2H), 5.40 (t, J = 6.28 Hz, 1H), 7.17 (s, 4H), 9.33 (s, 1H) ppm. ^13^C{1H} NMR (CDCl_3_): δ = 23.1, 23.3, 30.8, 34.68, 34.73, 45.8, 128.6, 128.8, 131.8, 131.9, 139.0, 140.4, 156.0, 156.6, 167.0 ppm. Elemental analysis, calcd (%) for C_26_H_38_N_2_O_5_S_2_ (522.72): C, 59.74; H, 7.33; N, 5.36; O, 15.30; S, 12.27; found: C, 58.53; H, 7.29; N, 5.30; O, 15.24; S, 12.12 (see Appendix A).

### 4.2. Synthesis of 1,2-Bis(4-(tert-butyl)-2,6-dimethylphenyl)disulfane (***3***)

Mixed toluene (30 mL) and glyoxal (0.3 g, 40%, 0.002 mol) were refluxed with distillation for 40–50 min until about 5–6 mL of the solvent remained in the flask. The mixture was then evaporated to dryness in a rotary evaporator in vacuo (70 °C bath temperature). To the residue in the flask were added titanium (IV) isopropoxide (6.5 mL) and sulfonamide **1** (1 g, 0.004 mol). The whole was then heated to 160 °C and stirred vigorously for 12 h. Upon completion, the reaction mixture was poured over with water (80 mL) and stirred vigorously for 1 h at room temperature. To the mixture was then added ethyl acetate (20 mL), followed by stirring for another 1 h. Upon completion, the mixture was filtered through a paper filter. The filter cake was washed with ethyl acetate several times. The extract was washed with water and brine, and dried over Na_2_SO_4_ and evaporated to dryness in a rotary evaporator in vacuo (50 °C bath temperature). The residue was subjected to preparative chromatography. Toluene was used as the eluent. Fractions with Rf = 0.76 were collected. The solvent was removed from the collected fractions in vacuo by a rotary evaporator (60 °C bath temperature) before the onset of crystallization. To the residue was added a small amount of acetonitrile, and the precipitation was allowed to finish. The suspension was filtered. The filter cake was washed with acetonitrile and dried at room temperature until constant weight. The result was compound **3** as a white crystalline powder.

Yield: 0.164 g (97.5% assay (HPLC)), 0.414 mmol (20% calculated as compound **1**). Mp = 125–127 °C. IR (KBr): ν = 2965, 2953, 2902, 2865, 1592, 1554, 1478, 1461, 1444, 1405, 1392, 1361, 1226, 1202, 1149, 1029, 997, 921, 870, 729, 633 cm^−1^. ^1^H NMR (CDCl_3_): δ = 1.31 (s, 18H), 2.24 (s, 12H), 7.05 (s, 4H) ppm. ^13^C{1H} NMR (CDCl_3_): δ = 21.7, 31.2, 34.4, 125.1, 131.6, 142.9, 152.5 ppm. Elemental analysis, calcd (%) for C_24_H_34_S_2_ (386.66): C, 74.55; H, 8.86; S, 16.59; found: C, 74.43; H, 8.91; S, 17.02. No oxygen was found in the compound (see Appendix A). 

### 4.3. Synthesis of Bis(4-(tert-butyl)-2,6-dimethylphenyl)sulfane (***4***)

Sulfonamide **1** (1 g, 0.004 mol) was dissolved in H_2_SO_4_ (1.33 mL, 94%). The mixture was then heated to 60 °C, and glyoxal (0.3 g, 40%, 0.002 mol) was added portion-wise with stirring for 1–2 min. The whole was then stirred for 4 h, maintaining the same temperature. Upon completion, the reaction mixture was poured over with water (20 mL) and ethyl acetate-extracted. The extract was washed with water and brine, and dried over Na_2_SO_4_ and evaporated to dryness in a rotary evaporator in vacuo (50 °C bath temperature). The residue was subjected to preparative chromatography. Toluene was used as the eluent. Fractions with Rf = 0.74 were collected. The solvent was removed from the collected fractions by a rotary evaporator in vacuo before the onset of crystallization (60 °C bath temperature). To the residue was added a small amount of acetonitrile, and the precipitation was allowed to finish. The suspension was filtered. The filter cake was washed with acetonitrile and dried at room temperature until constant weight to give compound **4** as a white crystalline powder.

Yield: 0.195 g (98% assay (HPLC)), 0.538 mmol (26% calculated as compound **1**). Mp = 129–131 °C. IR (KBr): ν = 2963, 2903, 2867, 1596, 1553, 1477, 1467, 1444, 1434, 1375, 1360, 1301, 1228, 1201, 1150, 1038, 1022, 1000, 866, 720, 640, 624 cm^−1^. ^1^H NMR (acetone-d): δ = 1.28 (s, 18H), 2.21 (s, 12H), 7.13 (s, 4H) ppm. ^13^C{1H} NMR (acetone-d): δ = 21.2, 30.7, 33.9, 125.6, 130.9, 139.7, 149.9 ppm. Elemental analysis, calcd (%) for C_24_H_34_S (354.59): C, 81.29; H, 9.66; S, 9.04; found: C, 80.64; H, 9.66; S, 9.12. No oxygen was found in the compound (see Appendix A).

### 4.4. Synthesis of Protonated 4-(Tert-butyl)-2,6-dimethylbenzenesulfonic Acid (***6(H^+^)***) Monohydrate

Compound **8** (5 g, 0.021 mol) was added dropwise to H_2_SO_4_ (11 mL, 94%) with vigorous stirring for 1 h at a constant temperature of 25 °C. The whole was then vigorously stirred for 5 h. Upon completion, a saturated NaCl solution (8.5 mL) was poured into the mixture. The mixture was cooled to 10 °C, stirred for 15 min and filtered. The filter cake was transferred to a beaker into which diethyl ether (35 mL) was then poured. To the mixture was further added H_2_SO_4_ with vigorous stirring until the sediment was fully dissolved. The extract was removed by decantation. The residue was extracted twice again with diethyl ether (8 mL each), repeating the procedure. The combined extracts were dried over CaCl_2_ and evaporated to dryness in a rotary evaporator. The result was protonated acid **6(H^+^)** monohydrate as a white crystalline powder.

Yield: 1.7 g (95% assay (HPLC)), 6.203 mmol (20.1% calculated as compound **1**). Mp = 125–129 °C. IR (KBr): ν = 3059, 2980 br., 2963, 2867, 1817 br., 1671, 1596, 1559, 1477, 1458, 1404, 1360, 1226, 1081, 1015, 868, 802, 677, 640, 601 cm^−1^. ^1^H NMR (CDCl_3_): δ = 1.31 (s, 9H), 2.61 (s, 6H), 7.09 (s, 2H), 9.58 (br. s, 2H) ppm. ^13^C{1H} NMR (CDCl_3_): δ = 23.0, 31.0, 34.5, 127.8, 134.6, 137.7, 154.3 ppm. Elemental analysis, calcd (%) for C_12_H_21_O_4_S (261.36): C, 55.15; H, 8.10; O, 24.49; S, 12.27; found: C, 54.4; H, 7.94; O, 25.20; S, 12.39 (see Appendix A).

### 4.5. Synthesis of 1,3,5-Tris((4-(tert-butyl)-2,6-dimethylphenyl)sulfonyl)-1,3,5-triazinane (***9***)

Sulfonamide **1** (1 g, 0.004 mol) was dissolved in mixed H_2_SO_4_ (9.19 mL, 94%) and water (1.67 mL) at room temperature. Formaldehyde (0.347 g, 36%, 0.004 mol) was then added portion-wise to the mixture with vigorous stirring at room temperature for 2–3 min. The mixture was further stirred vigorously, poured over with water (50 mL), and stirred another 10 min. The suspension was filtered, and the filter cake washed thrice with water. The sediment was transferred into a beaker and muddled actively in mixed 5:1 vol.% ethanol: acetone (15 mL) at 60 °C for 20 min. The mixture was then cooled, filtered, washed twice with ethanol, and dried until constant weight to furnish compound **9** as a white crystalline powder.

Yield: 0.55 g (96% assay (HPLC)), 0.697 mmol (50.5% calculated as compound **1**). Mp = 235–23 7°C (dec.). IR (KBr): ν = 2966, 2906, 2869, 1594, 1556, 1478, 1462, 1407, 1385, 1334, 1226, 1175, 1146, 1049, 1034, 1014, 918, 870, 739, 721, 673, 638 cm^−1^. ^1^H NMR (CDCl_3_): δ = 1.32 (s, 27H), 2.48 (s, 18H), 4.71 (s, 6H), 7.13 (s, 6H) ppm. ^13^C{1H} NMR (CDCl_3_): δ = 22.8, 30.9, 34.7, 58.2, 128.5, 130.9, 140.7, 156.2 ppm. Elemental analysis, calcd (%) for C_39_H_57_N_3_O_6_S_3_ (760.08): C, 61.63; H, 7.56; N, 5.53; O, 12.63; S, 12.66; found: C, 61.17; H, 7.50; N, 5.51; O, 12.71; S, 12.68 (see Appendix A).

### 4.6. Synthesis of N,N’-(1,2-Bis((4-(tert-butyl)-2,6-dimethylphenyl)sulfonamido)ethane-1,2-diyl)diacetamide (***12***)

H_2_SO_4_ (80 mL, 94%) was added portion-wise to mixed acetonitrile (80 mL), water (3.8 mL), sulfonamide **1** (3 g, 0.012 mol) and glyoxal (0.902 g, 40%, 0.006 mol) for 2–3 min, maintaining the temperature below 15 °C. The whole was then heated to 30 °C and stirred for 5 h. Upon completion, the reaction mixture was poured over with water (400 mL) and extracted with ethyl acetate. The extract was washed with water and brine, and dried over Na_2_SO_4_ and evaporated to dryness in a rotary evaporator in vacuo (40 °C bath temperature). The residue was subjected to preparative chromatography. Mixed chloroform and gracious acetic acid in a volume ratio of 10:0.5 were used as the eluent. Fractions with Rf = 0.15 were collected. The solvents were evaporated to dryness from the collected fractions by a rotary evaporator in vacuo (60 °C bath temperature). The residue was recrystallized from mixed diethyl ether and acetone in a volume ratio of 4:1 and dried at room temperature until constant weight. The result was compound **12** as a white crystalline powder.

Yield: 0.28 g (96% assay (HPLC)), 0.435 mmol (7% calculated as compound **1**). Mp = 207–209 °C (dec.). IR (KBr): ν = 3354, 3184, 2966, 2870, 1665, 1594, 1519, 1479, 1460, 1406, 1373, 1316, 1279, 1227, 1175, 1148, 1077, 1052, 929, 898, 870, 655 cm^−1^. ^1^H NMR (DMSO-d6): δ = 1.25 (s, 18H), 1.28 (s, 6H), 2.54 (s, 12H), 5.12 (d, J = 7.36 Hz, 2H), 7.15 (s, 4H), 7.35 (br. s, 2H), 7.70 (d, J = 7.2 Hz, 2H) ppm. ^13^C{1H} NMR (DMSO-d6): δ = 22.4, 23.3, 31.1, 34.7, 60.8, 128.3, 135.4, 138.5, 154.4, 168.9 ppm. Elemental analysis, calcd (%) for C_30_H_46_N_4_O_6_S_2_ (622.84): C, 57.85; H, 7.44; N, 9.00; O, 15.41; S, 10.30; found: C, 57.67; H, 7.40; N, 8.93; O, 15.53; S, 10.26 (see Appendix A).

### 4.7. Synthesis of 5-(Tert-butyl)-1,3-dimethyl-2-tosylbenzene (***13***)

A mixture of sulfonamide **1** (1 g, 0.004 mol), toluene (27 mL, dewatered with zeolites) and TfOH (0.44 mL) was stirred at reflux for 5 h. The whole was then cooled, poured over with water (50 mL) and stirred vigorously for 15 min. The mixture was further separated on a separation funnel. The water layer was additionally extracted once with toluene (8 mL). The organic phases were combined, washed with water and brine, and dried over Na_2_SO_4_ and evaporated to dryness in a rotary evaporator in vacuo (70 °C bath temperature). The residue was subjected to preparative chromatography. Toluene was used as the eluent. Fractions with Rf = 0.18 were collected. The solvent was evaporated to dryness from the collected fractions by a rotary evaporator in vacuo (60 °C bath temperature). The result was compound **13** as a yellowish resin that crystallizes over time. 

Yield: 0.50 g (95% assay (HPLC)), 1.512 mmol (73% calculated as compound **1**). Mp = 92–95 °C. IR (KBr): ν = 2962, 2907, 2865, 1594, 1555, 1459, 1404, 1383, 1363, 1302, 1225, 1168, 1143, 1089, 1032, 1016, 926, 871, 810, 762, 728, 707, 696, 672, 631, 616 cm^−1^. ^1^H NMR (CDCl_3_): δ = 1.31 (s, 9H), 2.42 (s, 3H), 2.65 (s, 6H), 7.12 (s, 2H), 7.29 (d, J = 7.92 Hz, 2H), 7.72 (d, J = 8.08 Hz, 2H) ppm. ^13^C{1H} NMR (CDCl_3_): δ = 21.6, 23.2, 30.9, 34.6, 126.4, 128.6, 129.5, 134.2, 139.6, 140.6, 143.4, 155.8 ppm. Elemental analysis, calcd (%) for C_19_H_24_O_2_S (316.46): C, 72.11; H, 7.64; O, 10.11; S, 10.13; found: C, 73.40; H, 7.70; O, 10.28; S, 10.27 (see Appendix A).

### 4.8. Synthesis of 2,2′-Sulfonylbis(5-(tert-butyl)-1,3-dimethylbenzene) (***14***)

Mixed sulfonamide **1** (1 g, 0.004 mol), 1,2-DCE (13.4 mL, dewatered with zeolites) and TfOH (0.33 mL) were stirred at reflux for 5 h. The mixture was then cooled, poured over with water (50 mL), and stirred vigorously for 15 min. The mixture was further separated on a separation funnel. The water layer was additionally extracted once with toluene (8 mL). The organic phases were combined, washed with water and brine, and dried over Na_2_SO_4_ and evaporated to dryness in a rotary evaporator in vacuo (70 °C bath temperature). The residue was subjected to preparative chromatography. Fractions with Rf = 0.24 were collected. The solvent was evaporated to dryness from the collected fractions by a rotary evaporator in vacuo (60 °C bath temperature). The result was compound **14** as a yellowish resin that crystallizes over time.

Yield: 0.51 g (98% assay (HPLC)), 1.305 mmol (63% calculated as compound **1**). Mp = 157–159 °C. IR (KBr): ν = 2965, 2906, 2868, 1592, 1557, 1481, 1458, 1404, 1361, 1301, 1227, 1169, 1132, 1058, 1034, 1010, 928, 869, 802, 730, 661, 644 cm^−1^. ^1^H NMR (DMSO-d6): δ = 1.27 (s, 18H), 2.37 (s, 12H), 7.22 (s, 4H) ppm. ^13^C{1H} NMR (DMSO-d6): δ = 21.7, 31.1, 34.9, 128.8, 137.77, 137.83, 155.5 ppm. Elemental analysis, calcd (%) for C_24_H_34_O_2_S (386.59): C, 74.56; H, 8.86; O, 8.28; S, 8.29; found: C, 74.62; H, 8.90; O, 8.33; S, 8.36 (see Appendix A).

## 5. Conclusions

Thus, we examined in detail the acid-catalyzed condensation between 4-tert-butyl-2,6-dimethylbenzenesulfonamide (**1**) and glyoxal in a molar ratio of 2:1 in aqueous H_2_SO_4_, aqueous acetonitrile and acetone.

An irreversible transposition of the sulfonamide/glyoxal condensation product, 1,2-bis((4-(tert-butyl)-2,6-dimethylphenyl)sulfonamido)-1,2-ethanediol (**5**), was found to take place in aqueous H_2_SO_4_ and to be accompanied by the 1,2-hydride shift to afford 2-((4-(tert-butyl)-2,6-dimethylphenyl)sulfonamido)-N-((4-(tert-butyl)-2,6-dimethylphenyl)sulfonyl)acetamide (**2**). The starting 4-tert-butyl-2,6-dimethylbenzenesulfonamide (**1**) in H_2_SO_4_ underwent partial hydrolysis to 4-(tert-butyl)-2,6-dimethylbenzenesulfonic acid (**6**), which was further desulfated to 1-(tert-butyl)-3,5-dimethylbenzene (**7**). 

It has been shown for the first time that aldehydes may act as a reducing agent in the generation of disulfanes from aromatic sulfonamides, as is experimentally proved. 1,2-Bis(4-(tert-butyl)-2,6-dimethylphenyl)disulfane (**3**) was synthesized and probably originated from 4-tert-butyl-2,6-dimethylbenzenethiol (**8**) in the presence of atmospheric oxygen. In this case, the thiol was generated by the reduction of acid **7** with glyoxal. Bis(4-(tert-butyl)-2,6-dimethylphenyl)sulfane (**4**) was discovered to originate presumably from the Brønsted acid-catalyzed Friedel–Crafts reaction.

The condensation between sulfonamide **1** and formaldehyde consequently furnished 1,3,5-tris((4-(tert-butyl)-2,6-dimethylphenyl)sulfonyl)-1,3,5-triazinane (**9**) in a 50.5% yield.

Acetone was found to engage in a reaction with sulfonamide **1** under the acid-catalyzed conditions. Aqueous acetonitrile in H_2_SO_4_ underwent hydrolysis to the acetamide that reacted with diol **5** to give *N*,*N*’-(1,2-bis((4-(tert-butyl)-2,6-dimethylphenyl)-sulfonamido)ethane-1,2-diyl)diacetamide (**12**).

It was discovered that diimine could not be obtained by the condensation between sulfonamide **1** and glyoxal in the media most often used for the synthesis of structurally similar imines, in aprotic solvents at reflux in the presence of strong Lewis or Brønsted acids or dewatering agents such as titanium (IV) isopropoxide.

The Friedel–Crafts reaction between an aromatic sulfonamide and a benzene derivative was carried out for the first time. A new synthetic strategy has been proposed herein that can considerably shorten the stages in the synthesis of in-demand organic compounds of symmetric and asymmetric sulfones via the Brønsted acid-catalyzed Friedel–Crafts reaction, starting from aromatic sulfonamides and arenes activated towards an electrophilic attack.

The present study allows for the conclusion that aqueous H_2_SO_4_ (73–83% in the mixture), aqueous acetonitrile or acetone are not suitable media for the acid-catalyzed cascade condensation of sulfonamide **1** with glyoxal. The condensation in these media comes amid a large number of side products, some of which are formed irreversibly.

## Data Availability

No applicable.

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
