# Peer review of "Condensation of 4-Tert-butyl-2,6-dimethylbenzenesulfonamide with Glyoxal and Reaction Features: A New Process for Symmetric and Asymmetric Aromatic Sulfones"

_molecules, 2022, doi:10.3390/molecules27227793_

Round 1

Reviewer 1 Report

The present manuscript addresses the issue of synthesizing complex cage compounds such as aza- and oxaazaisowurtzitanes, the promising high-energy materials. The discussion section of the manuscript shows a good organic chemistry background of the authors. This study has discovered new processes and mechanisms of the acid-catalyzed condensation in the topic in question. The authors have established an untypical behavior of the understudied aromatic sulfonamides, highly activated by donor substituents. The sulfonamides were shown to be reduced by aldehydes and also to engage in a Friedel-Crafts reaction with arenes in the presence of Brønsted acids. The authors have managed to discover a new synthetic process for sulfones, the compounds that have been known for over 100 years and used extensively in various fields. The findings are original and very appealing for the experts in organic chemistry, especially for those dealing with the problems of the synthesis of caged derivatives of   aza- and oxaazaisowurtzitanes and sulfones. The conclusions are consistent with evidence and arguments presented, and address the main question posed. The overall work is of interest and well-written, and I do recommend publication after minor revision.

Specific comments:

1. The protonated acid in the text is designated as both 6+ and 6 (H+). To be consistent, please choose either.

2. A mistake on page 7 in the sentence: “The reactions performed in pure H2SO4 with no extra water afforded the highest yield of compound 2 from 4”. Judging from the whole paragraph, both of the compounds are formed rather than one form the other.

On the same page, the sentence “The generation of compound 12 corroborates that diol 6 originates also from the condensation between sulfonamide 1 and glyoxal”: the wrong designation of the diol (it should be 6 but not 5).

3. In the sentence on page 8, “The reaction in 1,2-DCE was faster than in toluene and was incomplete, which is likely due to TfOH reacting with toluene and escaping the reaction mixture”: it seems like the meaning of the sentence is conveyed incorrectly. Please check if any.

Author Response

The authors' response to Reviewer 1 has been uploaded as a PDF file.

Reviewer 2 Report

The manuscript was not well written. Honestly, I have not enjoyed while reading this manuscript. 

1. page4, line 151-154 paragraph should be corrected. This paragraph was mixed with other language.

2. In page 8, line 260-261, should be modified.

3. lot spacial errors was observed correct them.

Author Response

The authors' response to Reviewer 2 has been uploaded as a PDF file.

Reviewer 3 Report

In this manuscript, Paromov and coworkers discovered the condensation of 4-tert-butyl-2,6-dimethylbenzenesulfonamide with glyoxal and reaction features, which provided a new process for symmetric and asymmetric aromatic sulfones. The results are interesting, which may be of interest to the readers of Molecules. Therefore, I recommend a publication after minor revision. There are some typing issues to be corrected, such as Page 4, line 153-153, there are some unknown words.

Author Response

The authors' response to Reviewer 3 has been uploaded as a PDF file.

Round 2

Reviewer 2 Report

Dear Authors,

Thanks for the modification. Even though this work is not novel and does not meet the criterion of urgency and novelty expected for “Molecules” publications.